# Study on Bridge Displacement Monitoring Algorithms Based on Multi-Targets Tracking

Jiajia Wang and Guangming Li *

School of Mechanical, Electrical and Information Engineering, Shandong University, Weihai 264209, China; wangjiajiasduwh@163.com
* Correspondence: gmli@sdu.edu.cn; Tel.: +86-631-568-8338

**Abstract:** Bridge displacement measurement is an important area of bridge health monitoring, which can directly reflect whether the deformation of bridge structure exceeds its safety permission. Target tracking technology and Digital Image Correlation (DIC) are two fast-developing and well-known methods for non-contact bridge displacement monitoring in Digital Image Processing (DIP) methods. The former's cost of erecting detection equipment is too large for bridges with a large span that need to locate more multi-targets because of its tracking only one target on a camera while the latter is not suitable for remote detection because it requires very high detection conditions. After investigating the evolution of bridge displacement monitoring, this paper proposes a bridge displacement monitoring algorithm based on multi-target tracking. The algorithm takes full account of practical application and realizes accuracy, robustness, real-time, low-cost, simplicity, and self-adaptability, which sufficiently adapts the bridge displacement monitoring in theory.

**Keywords:** bridge displacement monitoring; digital image processing; multi-targets tracking

## 1. Introduction

Due to the frequent occurrence of structural constructions' safety accidents in recent years, the national property and people's lives have suffered tremendous losses. Hence, it is deeply recognized that real-time online monitoring and health assessment of structural constructions, especially those newly built, are very important. In view of the emergence of new structural constructions, long-span bridges, and complex system buildings, the health monitoring and safety assessment of structures should be paid more attention. Structural displacement, as a key index of structural safety evaluation, can directly reflect whether the deformation of bridge structure exceeds its safety permission. It can provide effective parameters for structural damage identification and health monitoring. Displacement monitoring can be applied to wood [1], bridges [2], dams [3], anti-lock brake systems [4], stay cables [5], and vibration analysis in vitro imaging systems [6], etc.

For structural construction like bridges, there are still a lot of contact displacement monitoring methods in use, such as acceleration sensor and displacement meter [7], Global Positioning System (GPS) [8], electronic total station [9], laser interferometer [10], and so on. Contact monitoring methods often need to install sensors on bridges, which makes the maintenance of sensors become very difficult in the later period because of bridges' often crossing the sea or rivers. In order to overcome these limitations, non-contact displacement monitoring methods have gradually emerged in recent years. Among them, the Digital Image Processing (DIP) method stands out for its high accuracy and simple installation of equipment. At present, the Digital Image Correlation (DIC) [1,4,5,11–22] method is the mainstream for bridge displacement monitoring using DIP.

DIC, also called digital speckle correlation, is to obtain the deformation information of the Region of Interest (ROI) from two digital images before and after deformation. The image before

deformation can be called the reference image and the image after deformation can be called the deformed image. The basic principle of DIC is to mesh the ROI in the reference image, treat the motion of each sub-region as a rigid motion, and calculate the correlation according to the pre-defined correlation function through a certain search method for each sub-region. In the deformed image, the location of the sub-region after deformation is the region whose cross-correlation coefficient with the sub-region in the reference image is the maximum, and then the displacement of the sub-region is obtained. By calculating all the sub-regions, the deformation information of the whole field can be obtained. The DIC method is welcomed by some scholars because it is easy to implement and its calculation accuracy is high. Despite its great advantages, the DIC method has to carry out more tedious data processing in order to obtain better measurement results. Its computational efficiency can not meet the requirements of fast measurement. In the process of a speckle image's acquisition and transmission, there are also many factors (such as signal noise, data missing, etc.) that affect the accuracy of measurement results. Therefore, how to better apply the DIC method to the actual measurement; there are still many research works to be further improved [4,11–22].

Target tracking [23–26], as its name implies, is to locate the position of a specific target in each frame of the image and generate the motion trajectory of the target. Setting a target board on the point which needs displacement monitoring, the displacement of the point can be known by the target board's displacement after processing the sampled image. The target board for sampling is designed with special geometric patterns, which Light Emitting Diode (LED) lights.

Initially, many scholars set the target board as a rectangular board with special square black-and white geometric patterns and obtain the center of a window by a special algorithm. This method has high monitoring accuracy of sub-millimeters under ideal conditions. However, the target has a directional characteristic that is, when using the target as the photographed object, each of its sides must be horizontal or vertical. If it is tilted, the precision of displacement monitoring will be reduced dramatically [25]. Another disadvantage of this method is that its calibration of the visual monitoring system designed is complex, leading to large calculation. To overcome this problem, Lu et al. [26] changed the pattern on the target board into a circle so that the precision would not be affected even if the target board was tilted. Moreover, the computation complexity of this method is much less than the above method. In [26], the method's cost of erecting detection equipment is too large for bridges with a large span that need to locate more multi-targets because of its tracking only one target on a camera.

In order to overcome the limitations of DIC, target tracking technology and other existing approaches, a novel algorithm based on specific target tracking technology is proposed in this article that can measure displacements of multi-targets simultaneously and be free of complex data processing. Experiments have been conducted to test the algorithm's various properties with images obtained by Photoshop (Adobe Photoshop CS6, Adobe Systems Incorporated, San Jose, California, USA). Results show that this method realizes accuracy, robustness, real-time, low-cost, simplicity, and self-adaptability, which sufficiently adapts the bridge displacement monitoring in theory.

The remainder of the paper is organized as follows. Section 2 discusses the key fundamentals of target tracking technology and DIC. The proposed algorithm is described in Section 3. The experimental results are analyzed in Section 4. Section 5 draws conclusions.

## 2. Digital Image Correlation

The basic principle of DIC is to mesh the ROI in the reference image, treat the motion of each sub-region as a rigid motion, and calculate the correlation according to the pre-defined correlation function through a certain search method for each sub-region. In the deformed image, the location of the sub-region after deformation is the region whose cross-correlation coefficient with the sub-region in the reference image is the maximum, and then the displacement of the sub-region is obtained. By calculating all the sub-regions, the deformation information of the whole field can be obtained.

In the DIC method, we need to determine a correlation coefficient function, which is a criterion to measure the matching degree between reference subset and target subset and then find out the

extremum of correlation coefficient by corresponding integral pixel displacement search method to extract displacement information. About the extremum, we take the maximum value in positive correlation coefficient function and the minimum value in negative correlation coefficient function. Different expressions of correlation coefficients have different performance and its performance will directly affect the search accuracy and computation efficiency of the DIC method. Two commonly used correlation coefficient functions, Zero Normalized Cross-Correlation (ZNCC) and Zero Normalized Sum of Squared Differences (ZNSSD), are presented in Equation (1) [27–29]:

$$
\begin{aligned}
C_{ZNCC} &= \sum_{i=-M}^{M} \sum_{j=-M}^{M} \left[ \frac{\{f(x_i,y_j)-f_m\}\{g(x_i',y_j')-g_m\}}{\Delta f \Delta g} \right], \\
C_{ZNSSD} &= \sum_{i=-M}^{M} \sum_{j=-M}^{M} \left[ \frac{f(x_i,y_j)-f_m}{\Delta f} - \frac{g(x_i',y_j')-g_m}{\Delta g} \right]^2,
\end{aligned}
\tag{1}
$$

where $f()$ and $g()$ mean the gray level intensity of a pixel for the reference and deformed images,

$$
f_m = \frac{1}{(2M+1)^2} \sum_{i=-M}^{M} \sum_{j=-M}^{M} f(x_i, y_j),
$$

$$
g_m = \frac{1}{(2M+1)^2} \sum_{i=-M}^{M} \sum_{j=-M}^{M} g(x_i', y_j'),
$$

respectively, and

$$
\Delta f = \sqrt{ \sum_{i=-M}^{M} \sum_{j=-M}^{M} [f(x_i, y_j) - f_m]^2 },
$$

$$
\Delta g = \sqrt{ \sum_{i=-M}^{M} \sum_{j=-M}^{M} [g(x_i', y_j') - g_m]^2 }.
$$

The advantages of this method are that the algorithm is simple and easy to realize, it is the whole field of view and the non-contact deformation monitoring, and the monitoring precision is high.

The disadvantages of this method are that it is a deformation monitoring method according to the speckle image of the structure. If the texture of the structure surface is good, the speckle image with a good effect can be obtained, and the results of deformation monitoring will also be good. However, when the texture of the structure surface is general or no obvious texture, it is necessary to make artificial spots, otherwise the displacement can not be monitored by this method. In addition, it can be seen from the principle that the essence of this method is to carry out a large number of repeated calculations, and the overall calculation is very large. If the image pixels are large, the correlation computation of the two images would be extremely large so it can not meet the requirements of monitoring speed at all.

## 3. Proposed Algorithm

The digital image correlation method can calculate the whole field displacement. If it is used in bridge displacement monitoring, the actual environment is not suitable and the amount of calculation is very large so it is difficult to meet the monitoring speed. The algorithm proposed in this paper improves the specific target tracking technology to make it more meet the needs of actual bridge displacement monitoring. A flowchart of the proposed algorithm is shown in Figure 1. It is mainly composed of eight modules. In each module, the algorithm in this paper is optimized to meet the real-time, robustness, and adaptability as far as possible under the premise of monitoring accuracy.

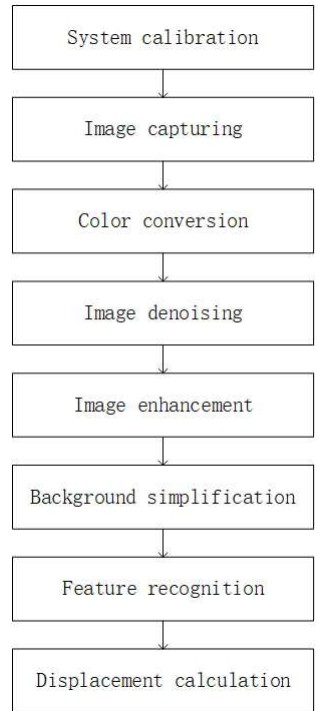

**Figure 1.** Flowchart of the proposed algorithm.

As mentioned earlier, current target tracking algorithms' cost of erecting detection equipment is too large for bridges with a large span that need to locate more multi-targets because of its tracking only one target on a camera, and DIC has difficulty guaranteeing the accuracy and real-time because of its complicated data processing. The proposed algorithm in this paper can meet the real-time and accuracy under the premise of monitoring multiple targets.

The algorithm proceeds as follows.

System calibration is acquiring the actual distance value represented by each pixel under the current monitoring distance before the displacement monitoring. The calibration method adopted in this paper is to separate two target boards by a known distance (preferably greater than 50 m), take photos, and run the algorithm to obtain the central coordinates of the two target boards in the image. Dividing the known distance by the coordinate difference between the two target boards is the actual distance respected by each pixel.

Image capturing means using the image acquisition system of the Charge Coupled Device (CCD) camera to collect images.

Color conversion means converting the collected images into gray images for processing.

Image denoising, as its name implies, is to filter out the noise in the image with a filter. In this paper, an adaptive median filter is used for denoising. $S_{xy}$ represents a sub-image whose center is at $(x, y)$. The adaptive median filter described in detail is as follows:

$z_{min}$ denotes the minimum brightness value in $S_{xy}$.

$z_{max}$ denotes the maximum brightness value in $S_{xy}$.

$z_{med}$ denotes the median brightness value in $S_{xy}$.

$z_{xy}$ denotes the brightness value at coordinates $(x, y)$.

This adaptive median filtering algorithm works in two levels, Level A and Level B:

Level A: If $z_{min} < z_{med} < z_{max}$, turn to Level B.

   Otherwise, increase window size.

   If the window size <=$S_{max}$, repeat Level A.

   Otherwise, output $z_{med}$.

Level B: If $z_{min} < z_{xy} < z_{max}$, output $z_{xy}$.

Otherwise, output $z_{med}$.

$S_{max}$ represents the allowable maximum adaptive filter windows size and $S_{max} = 5$ in this paper.

Many denoising methods are applied to this algorithm and the denoising effect is similar. In this paper, the adaptive median filter is selected and other denoising methods also can be used instead.

Image enhancement uses top-hat transformation to uniform background and histogram equalization to enhance the contrast. Assuming that $f(x,y)$ is a grayscale image and $b(x,y)$ is a structural element, $b$ erosion of image $f$ at $(x,y)$ is defined as Equation (2) and $b$ dilation of image $f$ at $(x,y)$ is defined as Equation (3). Top-hat transformation is defined as Equation (4):

$$[f \ominus b](x,y) = \min_{(s,t)\in b}\{f(x+s,y+t)\}, \tag{2}$$

$$[f \oplus b](x,y) = \max_{(s,t)\in b}\{f(x-s,y-t)\}, \tag{3}$$

$$T_{hat}(f) = f - (f \ominus b) \oplus b. \tag{4}$$

Background simplification uses erosion and hole filling technology. The structural element size of erosion procedure is a disk structure with a radius of 1. Hole filling technology is explained in Equation (5), where $A$ represents a binary image, $A^c$ represents the complementary set of $A$, $B$ represents a 4-connected structural element, and $X_0$ represents an all-black image except for a white spot in each hole. If $X_k = X_{k-1}$, the algorithm ends at step $k$ of the iteration. The parallel set of $X$ and $A$ is the image after hole filling:

$$X_k = (X_{k-1} \oplus B) \cap A^c, \quad k = 1,2,3,\dots \tag{5}$$

Feature recognition's flowchart is shown in Figure 2. The main principle is to take the vertices of eight directions on the edge of the region and calculate the distances from them to the center of the region, respectively. Then, the standard deviation of these eight distances is calculated and a threshold is set. If the standard deviation is less than this threshold, the region is recognized as a circle.

In this paper, the method of obtaining the center of a circle is the centroid location by first-order central moment method. It is a method of determining the centroid of image by using the geometric invariant first-order moment of a digital image. The concept of geometric moment was first proposed by Hu in 1962, and it is shown in Equation (6):

$$m_{pq} = \int_{-\infty}^{+\infty} \int_{-\infty}^{+\infty} x^p y^q f(x,y)dxdy, \tag{6}$$

where $f(x,y)$ means the continuous image function and $m_{pq}$ means $p+q$ order moments of the continuous image function.

For this paper, the image function $f(x,y)$ is a discrete numerical image so its integral operation is replaced by summation operation. $m_{pq}$ is shown in Equation (7):

$$m_{pq} = \sum_x \sum_y x^p y^q f(x,y). \tag{7}$$

The zero-order moment of the region represents the region's mass and the first-order moment represents the region's center of mass. The coordinates of the region's center of mass can be determined by combining the two, and it is presented in Equation (8):

$$\begin{aligned}
m_{00} &= \sum_x \sum_y f(x,y), \\
m_{10} &= \sum_x \sum_y xf(x,y), \\
m_{01} &= \sum_x \sum_y yf(x,y), \\
u_c &= m_{10}/m_{00}, \\
v_c &= m_{01}/m_{00},
\end{aligned} \tag{8}$$

where $(u_c, v_c)$ means the coordinate of the region's center.

Displacement calculation is to calculate the pixel displacement of the central coordinate of each circular target and determine the actual displacement distances according to the results of system calibration.

Compared with other image displacement monitoring algorithms, the algorithm in this paper has two main advantages. The first advantage is that the algorithm designs specific region detection conditions directly according to the characteristics of the target without image correlation method so it eliminates a lot of repeated calculations. When there are more pixels in the image, the advantage of this algorithm will be more obvious. The second advantage is that the algorithm can monitor multiple targets at the same time, unlike [25,26], where a camera can only monitor one target, which greatly reduces the cost of camera equipment. In summary, this algorithm is suitable for rapid displacement monitoring of large structures such as bridges.

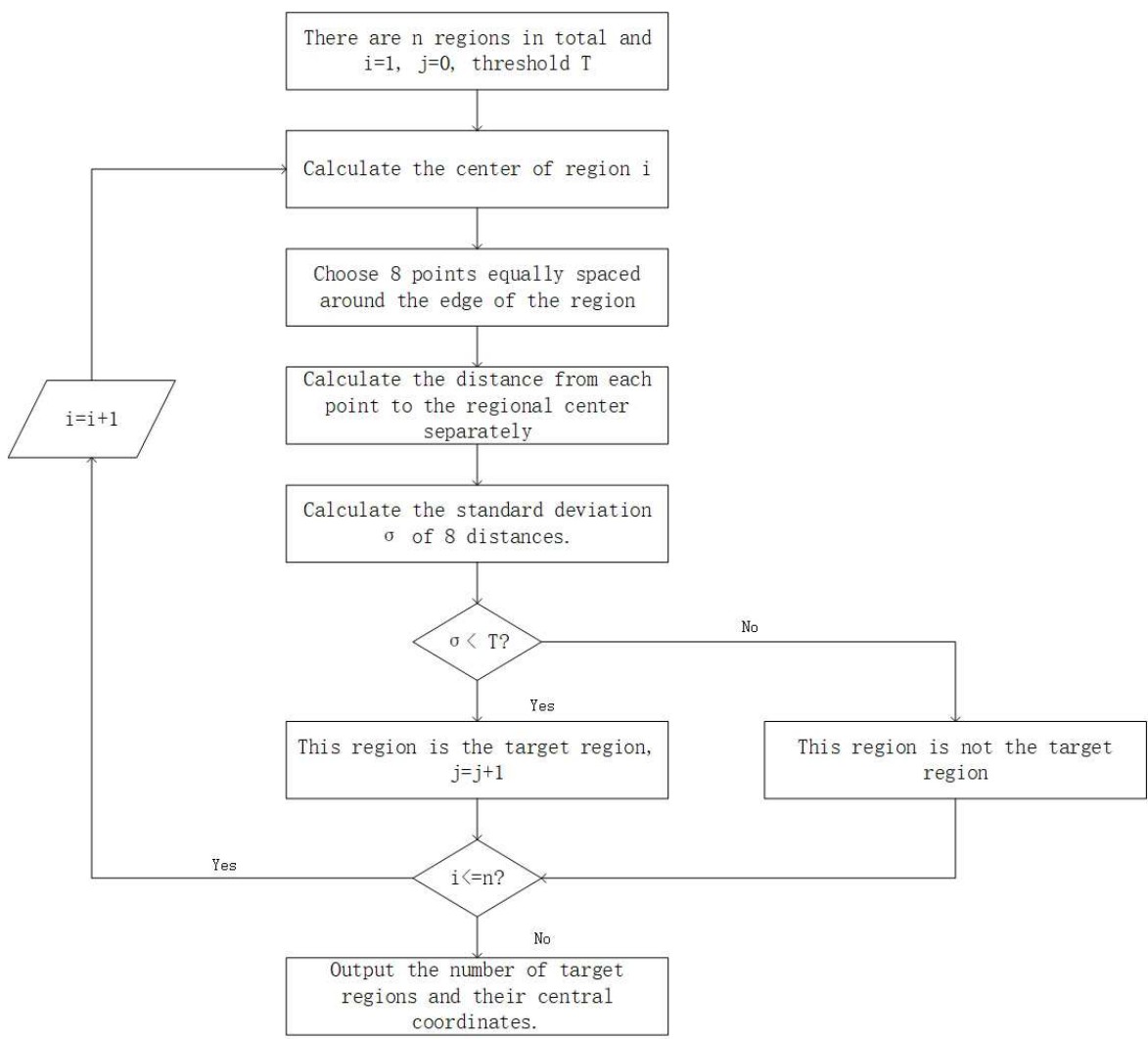

**Figure 2.** Flowchart of feature recognition.

## 4. Evaluation

In this paper, three experiments are conducted to verify the proposed algorithm's accuracy, robustness, real-time, self-adaptability, low-cost, and simplicity mentioned above. The original pictures in the experiments are made by Photoshop and the algorithm is realized by MATLAB language (R2016a, MathWorks, Natick, MA, USA).

In the first experiment, accuracy and self-adaptability can be proved. The accuracy of the algorithm can reach a sub-pixel level and meet the daily needs of bridges. This algorithm can automatically detect the number of target boards without setting a fixed number in advance, reflecting the self-adaptability of the algorithm. This algorithm realizes multi-target simultaneous monitoring and greatly reduces the cost of monitoring and facilitates later data processing, reflecting the low-cost and simplicity of the algorithm. In the second and third experiments, robustness and real-time can be proved respectively. The robustness of the algorithm is reflected in that, when the target board is slightly contaminated, the algorithm can still identify the location of the target board and the accuracy is almost unaffected. The algorithm is implemented in a frequency domain as far as possible and each step is calculated and compared to select the shortest time-consuming sub-algorithm from sub-algorithms that can achieve the same effect. If the displacement exceeds the threshold, real-time warning can be achieved. In the fourth experiment, the algorithm is achieved by C++ language to prove its availability in the actual environment.

*4.1. Accuracy Experiment*

As shown in Figure 3, we used Photoshop to make the picture to simulate actual scenes. We first move the target boards in the graph by certain amounts of displacement, which is shown in Table 1. The program calculates the target boards' displacement values of two pictures before and after the displacement. The results of the operation are shown in Figure 4 (The first column represents the *X* coordinate of the displacement, the second column represents the *Y* coordinate of the displacement, and each row represents the target board number), reflecting that the accuracy of the algorithm can reach a sub-pixel level.

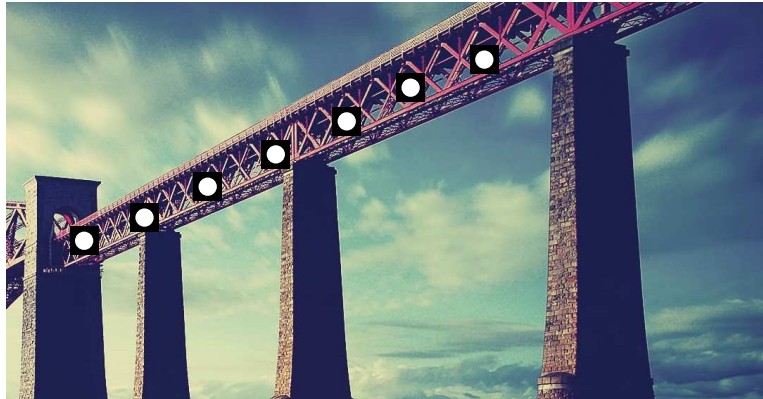

**Figure 3.** Image that simulates real environments (seven target boards).

**Table 1.** Certain displacement values of movement (the leftmost target board is No. 1).

| Target Board Number | Initial Position (Pixel) | Shifted Position (Pixel) | Displacement (Pixel) |
|:---:|:---:|:---:|:---:|
| 1 | (110, 339) | (117, 346) | (7, 7) |
| 2 | (169, 306) | (189, 306) | (7, 0) |
| 3 | (286, 262) | (286, 269) | (0, 7) |
| 4 | (383, 216) | (383, 209) | (0, 7) |
| 5 | (484, 169) | (491, 169) | (7, 0) |
| 6 | (575, 121) | (568, 128) | (7, 7) |
| 7 | (680, 81) | (680, 81) | (0, 0) |

|   | 1 | 2 |
|---|---|---|
| 1 | 6.9506 | 7 |
| 2 | 7.0246 | 0.0075 |
| 3 | 0 | 7 |
| 4 | 0.0246 | 6.9925 |
| 5 | 6.9260 | 0.0075 |
| 6 | 6.9754 | 7.0075 |
| 7 | 0 | 0 |

**Figure 4.** The displacement calculation results of the program.

To verify the self-adaptability of the algorithm, we replaced the original picture and repeated the experiment above. The image that simulates real environments is shown in Figure 5. The displacements set is shown in Table 2. The operation results of the program are shown in Figure 6.

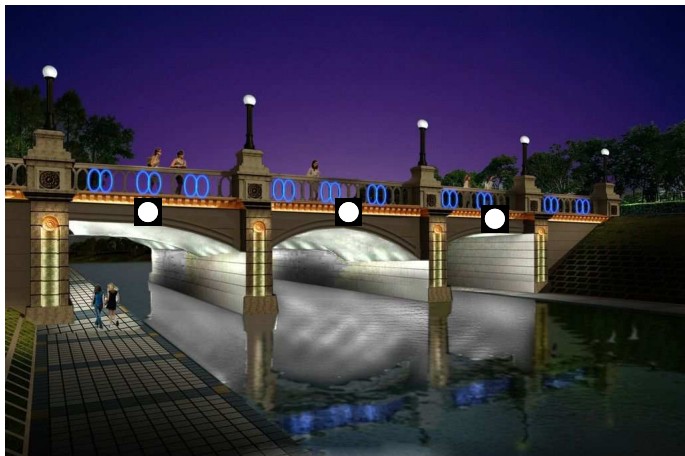

**Figure 5.** Image that simulates real environments (three target boards).

|   | 1 | 2 |
|---|---|---|
| 1 | 10 | 4.9794 |
| 2 | 10.0198 | 5.0103 |
| 3 | 4.9802 | 5.0103 |

**Figure 6.** The displacement calculation results of the program.

**Table 2.** Certain displacement values of movement (the leftmost target board is No. 1).

| Target Board Number | Initial Position (Pixel) | Shifted Position (Pixel) | Displacement (Pixel) |
|---|---|---|---|
| 1 | (214, 311) | (224, 306) | (10, 5) |
| 2 | (516, 311) | (506, 306) | (10, 5) |
| 3 | (736, 321) | (741, 316) | (5, 5) |

The results of the two experiments show that the accuracy of the proposed algorithm can reach 1/10th of a pixel and self-adaptability has also been proved very well.

### 4.2. Robustness Experiment

To verify the robustness of the algorithm, some stains of different shapes are added to the target boards in Figure 3. Pictures before and after displacement are shown in Figures 7 and 8. In order to reflect the advantages of the algorithm, we set different stains between pictures before and after displacement.

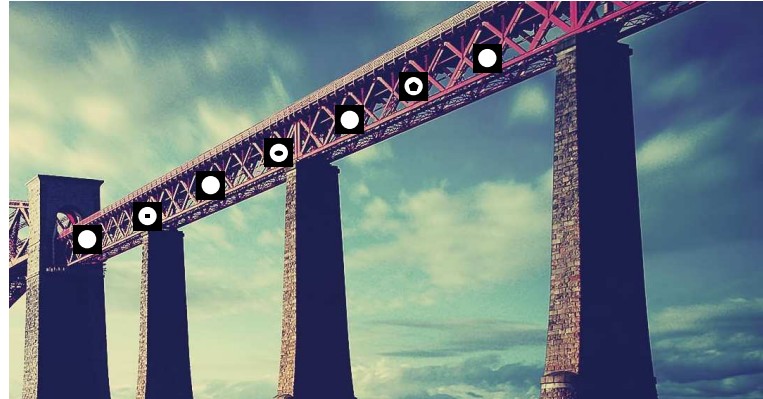

**Figure 7.** Image with stains before displacement.

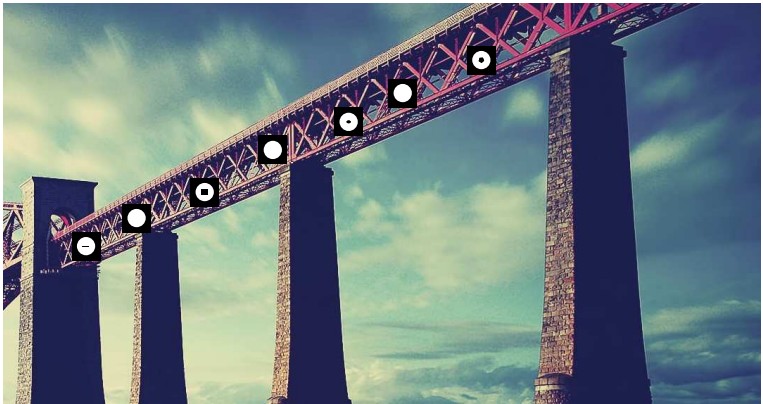

**Figure 8.** Image with stains after displacement.

Positions of target boards before and after displacement are shown in Table 1. The operation results of the program are shown in Figure 9. The experiment can prove the robustness of the algorithm and the accuracy is almost unaffected.

|   | 1 | 2 |
|---|---|---|
| 1 | 6.9754 | 7.0075 |
| 2 | 7.0247 | 0.0075 |
| 3 | 0.0494 | 7 |
| 4 | 0.0494 | 7 |
| 5 | 6.9754 | 0.0075 |
| 6 | 6.9754 | 7.0075 |
| 7 | 0 | 0 |

**Figure 9.** The displacement calculation results for images with stains of the program.

Figures 10 and 11 show the main process of image processing by the algorithm. The binarization threshold is 0.8 in this paper.

The original image. The image after color conversion.

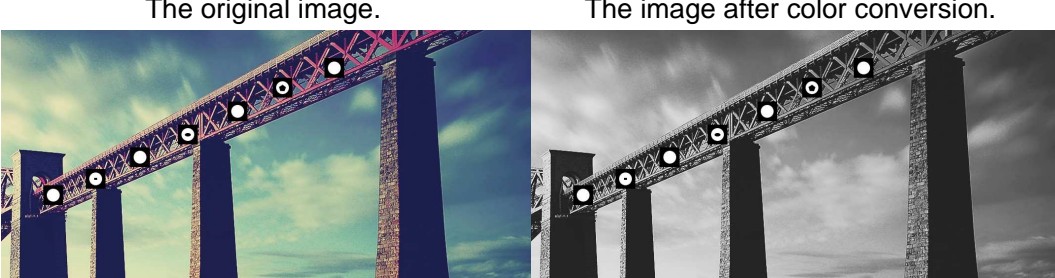

The image after top-hat transformation. The image after histogram equalization.

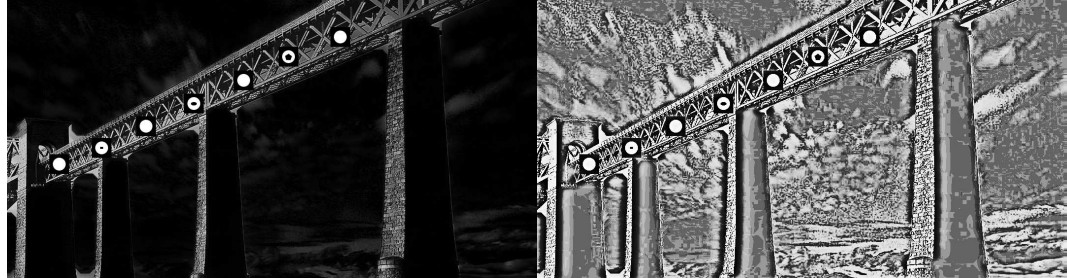

**Figure 10.** The main process of image processing by the algorithm.

The image after adaptive median filter. The image after binaryzation.

The image after hole filling. The image after erosion.

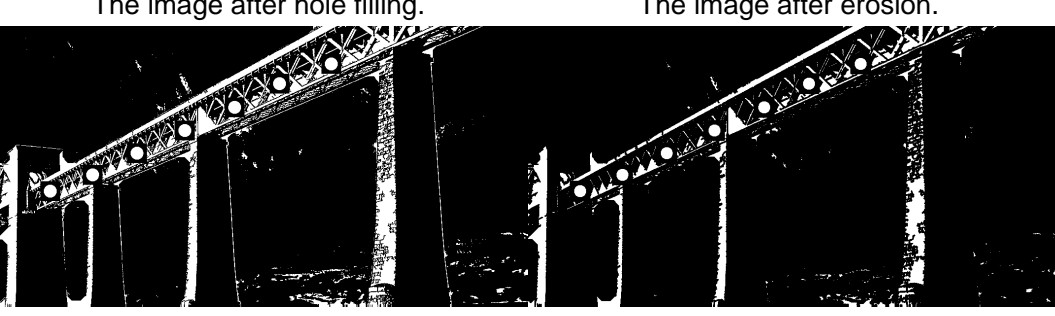

**Figure 11.** The main process of image processing by the algorithm.

### 4.3. Real-Time Experiment

In this experiment, three kinds of pictures in Figures 3, 5 and 7 are used to verify the real-time performance of the algorithm, respectively. Figures 3, 5 and 7's five sets of running time of the program are shown in Table 3. The results show that the real-time performance of the algorithm can achieve timely early warning.

**Table 3.** The running time of the program.

| Image | Running Time (s) |
|---|---|
| Figure 3 | 2.74, 2.71, 2.71, 2.73, 2.73 |
| Figure 5 | 3.29, 3.16, 3.14, 3.17, 3.17 |
| Figure 7 | 2.69, 2.70, 2.70, 2.69, 2.69 |

### 4.4. Actual Environment Experiment

In this experiment, the algorithm in this paper was achieved by C++ language. We bought the CCD camera and the turntables and built a real-time displacement monitoring system.

The experimental principle is shown in Figure 12. The large circle is a turntable and its background can be any. The right side of the turntable is the target board. Assuming that the distance between the center of the target board and the center of the turntable is $r$, the turntable is divided into 16 equal parts according to the angle so each angle is $\pi/8$. Assuming that the rotation angle is $\theta$, the starting position is shown in Figure 12 and the turntable is rotated in a counterclockwise direction. Assuming that the center of the turntable is the origin of the coordinate, the $x$-axis is the horizontal direction and the $y$-axis is the vertical direction, the coordinate of the center of the target board is $(r\cos\theta, r\sin\theta)$. The initial coordinate of the center of the target board is $(r,0)$ so the theoretical displacement of the target board rotated for one circle with the turntable is $(r - r\cos\theta, |r\sin\theta|)(0 \le \theta < 2\pi)$. In this experiment, $r = 500$ mm, the number of monitoring targets is 2, the radius of the target boards is 70 mm, the displacement monitoring distance is 10 m, the parameters of the CCD camera are shown in Table 4, and the results of the real-time displacement monitoring experiment are shown in Table 5. According to the calculation, the average error of the experiment is 3 mm, which verifies the effectiveness and reliability of the proposed algorithm. The CCD camera equipment is shown in Figure 13, and the turntable equipment is shown in Figure 14.

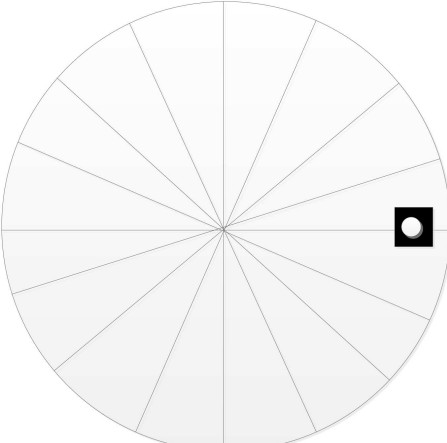

**Figure 12.** The experimental principle.

**Table 4.** The parameters of the CCD camera.

| Parameter Name | Parameter Size |
|---|---|
| Basler ace classic | acA2040-90uc |
| Sensor | CMV4000 |
| Resolution (H×V pixels) | 2048×2048 |
| Frame rate (fps) | 90 |
| Mono/color | c |
| Bit depth | 8/12 |
| Interface | USB 3.0 |
| Pixel size ($\mu m^2$) | 5.5×5.5 |
| Sensor size ($mm^2$) | 11.26×11.26 |
| Optical size | 1" |

**Table 5.** Results of the real-time displacement monitoring experiment.

| Theoretical Displacement (mm) | Monitoring Displacement (Left) (mm) | Monitoring Displacement (Right) (mm) |
|---|---|---|
| (0, 0) | (1.23, 0.02) | (0.08, 0.05) |
| (38.05, 191.35) | (39.78, 191.01) | (39.07, 190.89) |
| (146.45, 353.55) | (141.5, 354.05) | (142.3, 354.12) |
| (308.64, 461.95) | (303.52, 464.05) | (304.51, 463.50) |
| (500, 500) | (496.01, 502.27) | (498.24,502.35) |
| (691.35, 461.95) | (685.12, 465.10) | (684.66, 460.95) |
| (853.55, 353.55) | (847.48, 357.24) | (848.14, 351.46) |
| (961.95, 191.35) | (956.79, 194.02) | (957.03, 186.0) |
| (1000, 0) | (994.25, 0.91) | (994.28, 1.78) |
| (961.95, 191.35) | (956.26, 187.48) | (956.79, 194.12) |
| (853.55, 353.55) | (845.54, 353.89) | (847.48, 357.24) |
| (691.35, 461.95) | (684.40, 461.15) | (684.61, 460.62) |
| (308.64, 461.95) | (305.62, 463.25) | (305.51, 463.23) |
| (146.45, 353.55) | (142.50, 354.02) | (142.32, 354.08) |
| (38.05, 191.35) | (39.23, 191.05) | (39.01, 191.89) |

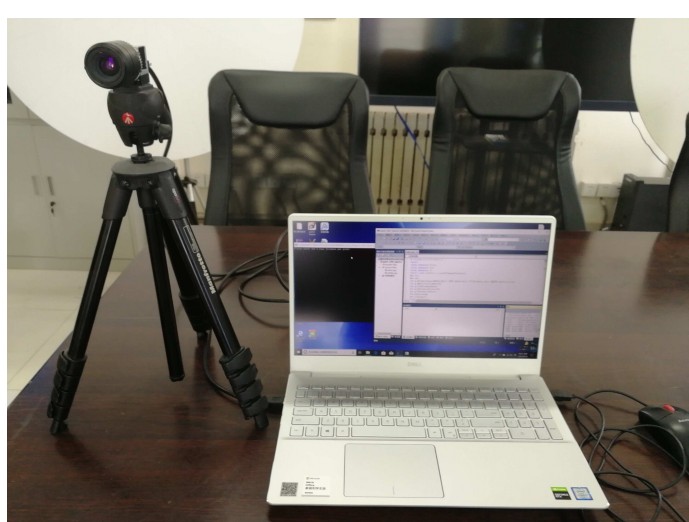

**Figure 13.** The CCD camera equipment.

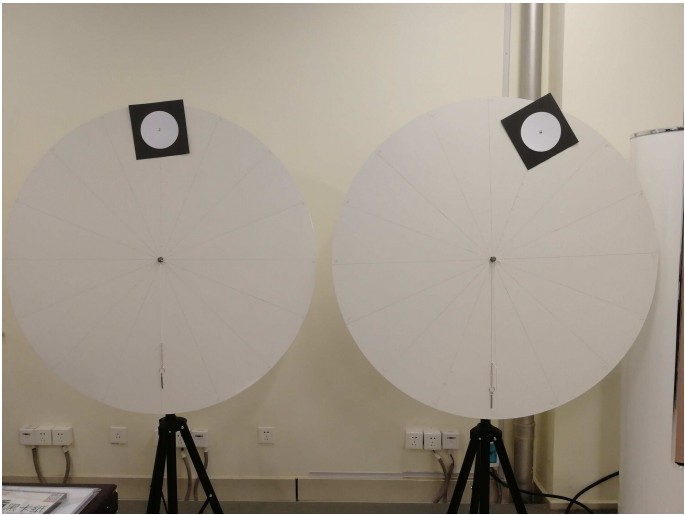

**Figure 14.** The turntable equipment.

## 5. Conclusions

This paper presents a novel displacement monitoring algorithm for real-time displacement monitoring of multiple targets. The proposed algorithm overcomes the unicity of tracking targets in single target tracking algorithms and the non-real-time performance in the DIC method. This paper sets three different experiments to verify the algorithm's performance and the results show that the proposed algorithm can realize accuracy of 1/10th of a pixel, robustness, real-time, low-cost, simplicity and self-adaptability. This paper also sets an actual environment experiment, and the results show that this method in the paper can achieve the displacement monitoring accuracy of 3 mm error at a 10 m monitoring distance. However, this paper focuses on the research of the algorithm and the preliminary construction of the real-time displacement monitoring system. This paper only verifies that the algorithm is effective for the displacement monitoring. The present research is not perfect, and the follow-up is still to be done. For example, the most remote monitoring distance is based on the camera resolution and the size of the tracking target board. What is the change law? What are the real issues to be overcome to use this system for the actual bridge?

**Author Contributions:** Writing—original draft preparation, J.W.; writing—review and editing, G.L. All authors have read and agreed to the published version of the manuscript.

**Funding:** The research was funded by Technology Developing Project of Shenzhen, Key Coordinative Innovation Plan of Guangdong Provice and Science & Technology Development Plan of Weihai.

**Acknowledgments:** The authors would like to thank Chengyou Wang, Fabao Yan and Ruijuan Jiang for their help and valuable suggestions. The authors would also be grateful for the support from SHENZHEN Municipal Design & Research Institute Co., Ltd. The authors also thank the anonymous reviewers and the editor for their valuable comments to improve the presentation of the paper.

**Conflicts of Interest:** The authors declare no conflict of interest.

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
