# Peer review of "Study on Bridge Displacement Monitoring Algorithms Based on Multi-Targets Tracking"

_futureinternet, doi:10.3390/fi12010009_

Round 1
Reviewer 1 Report
In this revision, it is good that the experiment using a turntable has newly been added. However, it is not clear what the relationship is between this experiment and the actual bridge environment. For example, it might be said that the scale-reduced model of the bridge displacement roughly corresponds to the turntable rotation. The additional explanation is desirable.
Author Response
We would like to thanks the reviewer for the positive and constructive comments and suggestions. Your valuable advice has played a great role in the later research of this project in our laboratory. I will feedback your comments to the next group of students who will take over the project. We have modified the paper according to the comments and added some contents which better explained the algorithm in this paper. Thanks again for your comments and valuable suggestions.

Reviewer 2 Report
The topic is interesting; however, many issues have to be addressed before a possible acceptance.
Which are the daily needs of bridges (line 162)? How is your method tested in those needs and actual environment conditions (line 11)?
Include references for the information presented in line 25
Check the text “...in Eq. 2…” in page 3.
Indicate the differences between the ZNCC and ZNSSD methods.
Why is the adaptive median filter used? There are other methods.
For readers, explain briefly the top-hat transformation.
It is not clear how the proposed method determines automatically the number of targets. Please be more descriptive, e.g. are the targets found from left to right? Up to down? What is the maximum number of targets? Minimum separation? How is the accuracy degraded according to the separation? How does the proposed method determine n in Fig. 1? How do you determine the threshold value? More tests are needed.
The settings for the proposed method are not mentioned; for instance, the filter size, the structural element size in the erosion procedure, etc. How did you determine such values? How many times the erosion procedure is carried out for an unknown image (provide results)? Your method has to be perfectly and totally reproducible.
The usefulness and performance of the DIP steps presented in Fig. 1 are no evident. Include in different figures the evolution of the input image according to the step carried out for the three study cases.
For the first two study cases, the size of figures and targets are not mentioned. Are they important? How do they impact in your proposal? The images of Figs. 3 and 5 represent unrealistic situations in terms of size for the targets; it is clear that the goals (accuracy and adaptability) are different but the size of the targets according to bridge size help to reduce problems of mix of textures. Include results with a single target but with a real area of a bridge in terms of size/texture.
The stains presented in Fig. 7 are unrealistic situations; although, different sizes and shapes are used, they are binary in color (black and white). Was the image binarized previously? Include results for noisy targets (where the automatic image thresholding can fail), even with no perfect squared shapes. The robustness cannot be claimed with the tests that have been carried out.
What do you mean with “three kinds of pictures” in line 196? Include them in the article. How do you determine that the values of table 3 satisfy real time? In this test, a velocity profile for the target has to be set, then a calculated profile using your proposal has to be compared. If the processing time is higher than the velocity profile, the real time cannot be achievable. As velocity profile, typical movements during seismic activity or ambient vibrations have to be considered. Also discuss the target velocity issue for section 4.4.
Why are two software (Matlab and C++) used in the different study cases?
In line 113, it is suggested distances greater that 50 m, but in the study cases the proportional size is not considered. In fact, the ratio between a distance of 10 m and the targets of 70 mm is not a realistic condition, small targets have to be considered in order to claim accuracy, sensibility, reliability, effectiveness, etc., mainly considering that the texture, size and distance of your turntable help to detect the target.
In order to claim simplicity and low-cost for your method, a comparison with other methods is required.
Re-write conclusions according to the changes carried out.
Author Response

(The authors gave the same response as above.)

Round 2
Reviewer 2 Report
The Reviewer really appreciates the work carried out to answer and follow his comments and suggestions. Although the manuscript is clearer and more complete, there is still some issues that have to be addressed.
The Reviewer understands that any research is perfect, there is always something to improve; in this regard, your method is not perfect either and does not require to have accuracy, robustness, real-time, low-cost, simplicity and self-adaptability to be considered as a great contribution, mainly considering that all your claims have to be clearly demonstrated, e.g., your results do not demonstrate real time operation (please check the term real time and consider if the online term is more appropriate).
Regarding answer 1: If your method is not tested under actual environment conditions, the sentence “which sufficiently adapts the displacement monitoring in actual bridge environment.” and other similar sentences of your work have to be removed.
Regarding answer 5: Such information has to be included and discussed in the manuscript. The idea is to inform to the readers the reason of your decision/choice, not to inform to the Reviewer. Please also check other answers.
Regarding answer 7: In the manuscript, the sentence “The threshold value is determined according to the actual environment.” is not stated. Again, how is the threshold value determined? If it is not determined in an automatic way, your method cannot be considered as adaptive.
Regarding answer 8: The median filter size is not mentioned. It is not clear how the application of only one erosion can delete the stains of different size in your results. Please comment on this.
Regarding answer 9: Include captions to the subfigures to indicate clearly the step performed.
Regarding answer 11: Include information to describe the hole filling technology. In which step is the image thresholded (is the Otsu algorithm used)?
Regarding answer 12: Anything about real time is provided. Again, the performed test and the obtained results do not demonstrate the real-time capacity. In the experiments presented in sections 4.3 and 4.4, anything about the velocity is said; in this regard, how can you discuss real time performance (it is impossible)? Online monitoring system is different to real-time monitoring system.
Author Response
We would like to thanks the reviewer again for the positive and constructive comments and suggestions. Your valuable advice has played a great role in the later research of this project in our laboratory. We have modified the paper according to the comments and added some contents which better explained the algorithm in this paper. Thanks again for your comments and valuable suggestions.

Round 3
Reviewer 2 Report
All my comments and suggestions have been addressed. I recommend the manuscript acceptance.
This manuscript is a resubmission of an earlier submission. The following is a list of the peer review reports and author responses from that submission.
Round 1
Reviewer 1 Report
This paper proposes an image processing algorithm for monitoring displacements in each location of the bridge. However, the verification experiments shown in the paper are limited to application to the synthetic images in which the known displacements are artificially added using Photoshop. Despite being stated in the paper title, no verification to the bridge image accompanied by ACTUAL (not artificial) displacements is performed. So the validity of the proposed method is unknown. A drastic revision of the experimental chapter is needed.
Reviewer 2 Report
This paper explore the well-known approach, named "digital image correlation", to measure the bridge displacement. Within the multi-target tracking framework, the authors claim the effectiveness of proposed approach. The paper is quite starghtforword, and easy to follow. However, my main concern is the novelty of this paper as the technical contribution is quite low. Moreover, the related work is not well presented.
Reviewer 3 Report
The paper focuses on the bridge displacement monitoring algorithms based on the multi-targets tracking. It is an interesting paper. However, the authors need to address the following issues before this manuscript is accepted:
In Section 2, the digital image correlation and target tracking methods are not presented by the authors. The relative description and references should be given. In Section 3, the flowchart of the proposed algorithm should be revised. In current figure 1, the authors assumed that all steps can be correctly performed. However, some errors may be occur. If some errors are produced, the relative method should be provided. In Figure 3, the target board number should marked according to that in Table 1. The similar issues in Figures 5, 7, and 8 should be revised too. Figures 4, 6, and 9 can be removed. The results can be described in the text. In Section 5, the descriptions in Conclusion are too simple. More indepth descriptions should be given. The language in the paper should be improved.The rest of the paper is fine. I look forward to receiving a revised version, with all the changes made highlighted.